# Relationships between Physical Activity Parenting Practices and Children’s Activity Measured by Accelerometry with Children’s Activity Style as a Moderator—A Cross Sectional Study

**DOI:** 10.3390/children9020248

**Published:** 2022-02-12

**Authors:** Mahnoush Etminan Malek, Åsa Norman, Liselotte Schäfer Elinder, Emma Patterson, Gisela Nyberg

**Affiliations:** 1Department of Global Public Health, Karolinska Institutet, Tomtebodavägen 18A, 171 77 Stockholm, Sweden; asa.norman@ki.se (Å.N.); liselotte.schafer-elinder@ki.se (L.S.E.); emma.patterson@ki.se (E.P.); gisela.nyberg@ki.se (G.N.); 2Department of Clinical Neurosciences, Karolinska Institutet, 171 65 Stockholm, Sweden; 3Department of Psychology, Stockholm University, 106 91 Stockholm, Sweden; 4Centre for Epidemiology and Community Medicine, Region Stockholm, Solnavägen 1E, 113 65 Stockholm, Sweden; 5Section for Risk and Benefit Assessment, Swedish Food Agency, Box 622, 751 26 Uppsala, Sweden; 6The Swedish School of Sport and Health Sciences (GIH), Lidingövägen 1, 114 33 Stockholm, Sweden

**Keywords:** physical activity, parenting practices, parental support, structural equation modeling, activity style, child temperament

## Abstract

The objective was to examine the associations between physical activity parenting practices (PAPP) and children’s levels of moderate-to-vigorous physical activity (MVPA), and time spent sedentary (SED) during non-school time in weekdays and weekends when children’s activity style was taken into account. Study participants were 88 children (mean age 6.3 (SD 0.3) years); 51.0% girls) and their parents who took part in A Healthy School Start Plus in Sweden. The independent variables included PAPPs Structure, Neglect/control, and Autonomy promotion and children’s activity style as moderator, assessed through validated parent questionnaires. Dependent variables were the MVPA and SED in minutes, measured by accelerometry. Structural equation modeling was used to examine the associations between PAPPs and children’s MVPA and SED with children’s activity style as a moderator. No significant associations between the PAPPs Structure, Neglect/control, and Autonomy promotion and measures of physical activity were found (*p* > 0.13). The moderating role of activity style improved the model fit and the final model had a reasonable fit to the data. Our results suggest that in future studies, with the aim to explore the relationship between PAPP and children’s physical activity, the activity style of the children should be included as a moderator.

## 1. Introduction

Insufficient physical activity is a leading risk factor for non-communicable diseases [1]. It is known that physical activity is associated with many physical and mental health benefits, such as improved bone health, cardiorespiratory and muscular fitness, and metabolic and cardiovascular health biomarkers [2,3]. When physical activity behaviors are established at a young age, it is more likely that the behaviors are carried over into adulthood [4]. The World Health Organization (WHO) has adopted a global recommendation of 60 min of moderate to vigorous physical activity (MVPA) per day for children up to the age of 18 years [5]. In addition, vigorous-intensity activities should be included at least three times per week. When looking internationally, device-measured estimates of children’s MVPA suggest that merely 45% of children meet the daily recommendation [6,7]. Furthermore, it is recognized that a significant reduction in the level of physical activity occurs between the age of 6–12 years [6]. Therefore, the need for action at all levels of society to increase population levels of physical activity has been emphasized, especially in school-aged children.

Another serious public health concern is the increase of child obesity during the last decades [8,9,10]. Prevention and treatment early in life should remain a priority [11,12,13,14]. The literature suggests that interventions that focus on managing and preventing child obesity should focus on behavior change or promotion of healthy dietary intake and physical activity behavior [15,16], and that school-based interventions involving parents and combining physical activity and diet may be an effective strategy for childhood obesity prevention worldwide [15]. However, such interventions have had limited success on physical activity [17,18]. 

In order to define interventions that can successfully affect children’s physical activity, it is essential to understand what factors influence their physical activity behavior. Research highlights the importance of parenting practices on children’s participation in physical activity [19,20]; however, the nature and extent of this influence has not been clearly defined. Parenting practices are described as goal-directed strategies in specific situations, performed by the parent in relation to the child [21]. In the literature, specific parenting practices related to children’s physical activity have been described as, e.g., monitoring, praising, or encouraging a child’s physical activity, modeling physical activity, or pressuring the child to engage in physical activity [20]. Significant associations have been identified between parental role modeling and parental support of physical activity and children’s activity levels [20]. Studies using qualitative methods have also identified logistic support, providing accessibility to physical activity and parental role modeling of physical activity as facilitators of children’s physical activity. In contrast, lack of parental cooperation and negative parent–child interactions may act as barriers to physical activity [22]. 

One factor that could covary with parenting practices and, thus, influence children’s physical activity, is the characteristics of the child itself. There is a general acceptance that the characteristics of the child play a vital role in shaping their emotional environment [23]. Emotionality (the tendency to become aroused easily and intensely), activity level (preferred levels of activity and speed of action), and sociability (the tendency to prefer the presence of others to being alone) appear early in development and are among the most heritable and stable temperament traits [24]. The view of a child as being an active agent in their own upbringing has its roots in psychological research on temperament [24]. The child’s temperament with regard to activity level, in this study referred to as activity style, is included in the Emotionality Activity Sociability (EAS) Temperament Survey for Children: Parental ratings [24]. This variable is a potential moderator for the relationship between physical activity parenting practices (PAPP) and children’s physical activity. 

Given the limited knowledge regarding parenting practices on physical activity as well as potential moderators, such as children’s activity style, we sought to examine these association using structural equation modeling (SEM). SEM is a statistical technique used to measure and analyze the relationships of observed and latent variables. Similar but more powerful than regression analyses, it examines linear relationships among variables, while simultaneously accounting for measurement error [25]. We examined the associations between physical activity parenting practices (PAPP) and children’s physical activity, with children’s activity style as a moderating factor. It was hypothesized that the relationships between physical activity parenting practices (PAPP) and children’s physical activity would be moderated by children’s activity style. 

## 2. Materials and Methods

### 2.1. Recruitment

The participants of this study comprised children and parents participating in the baseline measurement of the Healthy School Start Plus (HSSP) intervention study in September–October 2017 [26]. The HSSP is a parental support program with the overall aim of promoting physical activity and healthy dietary behaviors and prevent unhealthy weight gain, in pre-school class children (5–7-year-olds), with a special focus on parenting practices and behavior. Schools in mid-Sweden, where fewer parents than the national average had a university education, were invited. In the 17 schools that accepted the invitation, all families with children in pre-school class were invited and 352 families consented to participate in the HSSP intervention study, which was a cluster-randomized trial. More details of the sampling can be found in the HSSP study protocol [26].

Participants included in the sample for this study comprised all parent–child dyads who had complete data regarding parental questionnaires (parenting practices concerning the children’s physical activity and the children’s activity style), and children with valid accelerometer registrations. Another inclusion criterion was that the same parent had answered both questionnaires on parenting practices and children’s activity style. 

### 2.2. Ethical Statement

Written consent was collected from the parents. The HSSP study was approved by the Regional Ethical Review Board in Stockholm (No. 2017/711-31/1).

### 2.3. Data Collection and Measures

#### 2.3.1. Children’s Physical Activity and Sedentary Time

The children’s physical activity and sedentary time were measured with accelerometers (GT3X+, Actigraph, LCC, Pensacola, FL, USA), which are considered reliable and valid instruments [27]. The children and their parents were instructed that the children should wear the accelerometers on the right hip for seven consecutive days during wake time except for activities involving water. To process the accelerometer data, ActiLife Data Analysis version 6.13.3 was used. The data were saved in 5 s epochs at a sample rate of 30 Hertz and non-wear time was defined as 60 min of consecutive zeros allowing for 2 min of non-zero interruptions. A time filter was set between 07.00 and 22.59 and a separate time filter was used for non-school time between 07.00 and 07.59 and 16.00 and 22.59. Children who registered ≥500 min of activity per day were included. Internationally agreed cut-points were used to estimate moderate to vigorous physical activity (MVPA) (≥2296 counts per minute) and sedentary activity (SED) (0–100 counts/minute) [28,29]. Time (minutes) spent in MVPA and SED were calculated for non-school time on weekdays and for weekend days. 

#### 2.3.2. Children’s Anthropometry 

Trained research assistants made all measurements. Body composition was measured according to standardized procedures, using SECA instruments. All measurements were taken twice with a level of precision of 100 g for weight and 1 mm for height [26]. The children’s weight status (underweight, normal weight, overweight, and obesity) was classified according to the International Obesity Task Force [30].

#### 2.3.3. Level of Education and Region of Birth

The level of education was self-reported in a questionnaire by the parents. The family was classified as “high level of education”, if at least one parent reported an education longer than 12 years, otherwise the family was classified as “low level of education”. The family was classified as “Nordic”, if both parents reported their country of birth as Sweden, Norway, Denmark, Finland, or Iceland; otherwise, they were classified as “non-Nordic”.

#### 2.3.4. Physical Activity Parenting Practices and Children’s Activity Style

The items used to assess parenting practices regarding children’s physical activity (PAPP) were based on the item bank developed by Masse et al. [31]. This item bank was initially tested in a sample of 134 Canadian and US parents of 5- to 12-year-old children but did not display satisfactory validity [32]. During the development of the questionnaires for the HSSP intervention, few instruments existed showing acceptable psychometric properties. However, a new item bank was at the time under development by 24 experts from 6 countries and we selected 25 items from that item bank. Since then, the final item bank and its good psychometric properties have been described [33] with reasonable to good model fit (root mean square error of approximation (RMSEA) < 0.08, the comparative fit index (CFI) ≥ 0.95, and internal consistency ranging from 0.79 to 0.94). Furthermore, items have been deemed invariant across groups of parents’ sex, ethnicity, and income [33]. Of the 25 items initially chosen from the first item bank for this study, 9 items were not included in the final item bank; therefore, 16 items were included in the present study.

All items were translated into Swedish and back translated by the research team and pre-tested for comprehension in parents with pre-school children.

Three different latent PAPP domains were included in the analysis: Structure, Neglect/control, and Autonomy promotion (Table 1). The domain Structure, which refers to structuring the child’s social and physical environment to promote physical activity participation [31], contained the constructs Nondirective support (items 1–5), Supportive expectation (item 6), and Restrict indoor PA (items 7 and 8, reversed). The domain Neglect/control refers to pushing or pressuring the child to be physically active without considering the child’s interest and neglecting to structure participation in physical activity [31]. It contained the construct Coercive control (items 9–12, reversed). The domain Autonomy promotion refers to supporting participation in physical activity by encouraging participation, promoting autonomy and acknowledging participation [31]. It contained the constructs Autonomy support (items 13 and 14) and Guided choice (items 15 and 16). The responses of all items were captured on a 5-point scale with endpoints from ‘never’ to ‘very often’. The total values for the PAPP domains differed; ranging from 0–40 for Structure, where eight items were included, and from 0–20 for Neglect/control and Autonomy promotion, where four items were included in each domain. Internal consistency, determined by Cronbach’s alpha, was found to be 0.60 for Structure, 0.76 for Neglect/control, and 0.57 for Autonomy promotion.

In this study, activity style was defined as the child’s temperament with regard to activity, i.e., the child’s automatic preference, attitude, and choice with regard to the activity level in activities as part of the child’s day to day life. The items used to assess the children’s activity style originate from the Emotionality Activity Sociability (EAS) survey consisting of four scales: the emotionality scale, the activity scale, the shyness scale, and the experimental sociability scale [24]. In order to keep the response burden on participants as low as possible, the three most suitable items for this study out of five from the activity scale, a measure of vigor and tempo, were picked: (1) My child is always on the go; (2) My child is very energetic; and (3) My child prefers quiet, inactive games to more active ones (reversed). The psychometric properties have been examined in both European and north European contexts, displaying good psychometric properties [23,34,35]. 

The chosen items were rated on a five-point scale ranging from 1 (do not agree) to 5 (agree fully). For the analysis, a mean value for the children’s activity style was calculated, ranging from 0–5. The internal consistency for activity style assessed by Cronbach’s alpha was 0.67. All questionnaires were web-based, in Swedish, and accessed by parents via the project website. 

#### 2.3.5. Data Analysis

To test for differences between the included and non-included participants in this study, an independent samples *t*-test was used for continuous data and a chi-square test for categorical data. 

SEM was employed to assess the fit of the hypothesized model to the data [36]. We tested our model, which predicts that the three latent PAPP domains Structure, Neglect/control, and Autonomy promotion have a direct relationship with MVPA and SED, and that this is moderated by the children’s activity style. The suggested final conceptual model is presented in Figure 1 and was developed and tested with guidance from a statistician.

Three steps were taken to reach the final model. In the first step (Model 1), a path model between the three PAPP domains and the children’s physical activity was estimated. In the second step (Model 2), the children’s activity style was added as a moderator. In the third step (Model 3), the model fit was optimized by the inclusion of correlations between error terms.

Model fit was assessed based on the four indices: chi-squares/degrees of freedom ratio, the root mean square error of approximation (RMSEA) with 90% confidence interval, comparative fit index (CFI), and standardized root mean residuals (SRMR) [37]. The index chi-square/degrees of freedom ratio is sensitive to sample size, although a good model fit is in general indicated by a ratio <2. Model fit for RMSEA: <0.05 indicates good model fit, <0.08 acceptable fit, and 0.08 to 0.10 mediocre fit. Model fit for CFI: ≥0.9 indicates acceptable fit, and 0.95 good fit. Model fit for SRMR: <0.1 indicates acceptable fit [37].

The statistical analyses were performed using the software IBM SPSS, version 27.0 (Chicago, IL, USA), and model testing was conducted in IBM SPSS AMOS, version 27.0. The level of significance was set to *p* < 0.05.

## 3. Results

### 3.1. Descriptive Statistics

Table 2 presents the descriptive characteristics for the included children and the parents. In total, 88 parents (69 mothers and 19 fathers) and 88 children were included. Of the parents, 35% had finished 12 years or less of schooling and 32% were born outside the Nordic region. The mean age of the children were 6.3 (SD 0.3) years; 49% were boys (*n* = 43) and the rate of overweight and obesity among the children was 23%. Overall, the children spent on average 60 (SD 2.4) minutes in MVPA during weekends and 22 (SD 9.4) minutes during non-school time on weekdays. Of the children, 78% met the daily recommendation of 60 min MVPA per day. They spent on average 416 min in sedentary time during weekends and 167 min during non-school time in weekdays. Two significant differences between the participants included in this study and the rest of the HSSP participants were detected. The proportion of families born outside the Nordic region was lower (32% vs. 59%, *p* < 0.001) and the time spent sedentary during non-school time was higher (167 vs. 152 min, *p* = 0.014) in the participant group. 

### 3.2. Model Development and Activity Style as Moderator

The fit indices for the three different models are displayed in Table 3. Model 1 did not fit the data well, as the only acceptable index was the chi-square/degrees of freedom ratio. The fit was somewhat improved in Model 2, where activity style was added as a moderator. All indices displayed an improved fit. In Model 3, where error terms were correlated (Figure 1), the indices were even better, showing an acceptable fit based on the chi-square/degrees of freedom ratio and RMSEA. 

In Table 4, the relationships between the PAPP domains and children’s physical activity in Model 1 and Model 2 are displayed. None of the associations between the PAPP domains and children’s physical activity were significant (all *p* > 0.05). Children’s activity style did not significantly affect the children’s physical activity (*p* > 0.26). 

### 3.3. PAPP Domains and Children’s Activity Style

Table 5 shows results from Model 3 where the relationships between the three PAPP domains and the children’s activity style on children’s time spent in MVPA and SED is shown. There were still no significant associations between any of the PAPP domains or the children’s activity style with the children’s physical activity (*p* > 0.05), despite the improved model fit. Positive but non-significant associations were found for non-school time MVPA during weekdays with all three PAPP domains, and in general, negative non-significant associations were indicated for SED time on weekends. The direct effect of the manifest variable Activity style, which was a moderator in the model, indicated a positive but non-significant association with MVPA during non-school time on weekdays and during MVPA weekends and negative non-significant associations with SED non-school time during weekdays and SED weekends *(p* > 0.22).

## 4. Discussion

The purpose of this study was to examine the associations between the physical activity parenting practices (PAPP) Structure, Autonomy promotion, and Neglect/control, and children’s physical activity measured with accelerometry while considering a moderating effect of the children’s activity style. To our knowledge, this is the first study exploring the effect of a child’s physical activity style as moderator between PAPP and a child’s physical activity, using items from the updated item bank developed by Masse et al. [33]. Although no significant associations were found in our analyses, our conceptual model showed acceptable fit to the data, suggesting that activity style should be further explored as a moderator when analyzing the relationship between PAPP and children’s physical activity. 

When activity style was added as a moderator in our model, the model fit was improved, and although no significant associations between the PAPP domains with the children’s physical activity were found, this might indicate that the activity style of the child is a factor that should be considered when modeling the relationship between PAPP and children’s physical activity. In another study where the relationships between PAPP and child and family and environmental factors in families were examined, it was suggested that a child’s temperament influences PAPP [38]. In that study, the temperament was assessed by using the Colorado Childhood Temperament Inventory [39], where the parent’s perception of six factors in the child’s temperament was evaluated, namely sociability, emotionality, activity, attention span-persistence, reaction to food, and soothability. The authors suggested that the frequency of physical activity support from parents is influenced by the child’s temperament [38]. These findings strengthen our proposed need for further investigations of the role of children’s activity style when examining the relationship between PAPP and children’s physical activity, even though we only included the children’s activity style as an indicator of temperament as a moderator in this study. 

In the Koala Birth Cohort Study from the Netherlands, the authors examined whether a child’s background characteristics (BMI, gender, birth weight, eating style, and activity style) moderated the impact of the parenting practices [40]. Whether the child had an active activity style was assessed by not previously validated items: Compared to peers; (1) my child is very active; (2) never sits still. However, in this study, no interactions were found between activity-related parenting practices and the child’s activity style.

The direct effect of the activity style in our final model indicated a positive (non-significant) association with time spent in MVPA and a negative association with SED during non-school times on weekdays and during weekends. In a study by Song et al., it was examined whether the temperament activity level (assessed with different items from a different questionnaire than used in our study) in young children was associated with MVPA, and if parenting behaviors moderated these associations [41]. It was found that higher temperament activity level in early childhood predicted higher MVPA in later childhood and adolescence. The authors concluded that for children whose temperament activity level was low, parental support for physical activity may be beneficial [41]. In line with this, in a study focusing on four dimensions of temperament with the potential to influence physical activity in youth, it was concluded that childhood temperament, especially in males, may influence the development of future physical activity habits [42].

We could not confirm an effect of PAPP on young children’s physical activity during non-school time in our study. A meta-analysis examining parental correlates and potential moderators of children’s physical activity showed that the only support behavior with a moderate effect size was the relationship between parental encouragement and child PA (r = 0.34, 95% CI 0.25–0.41) [43]. The findings demonstrated that both parental support and modeling were related to child and adolescent physical activity. The authors highlighted that further research is needed on the role of others outside the family unit and that it may be important to consider how children’s siblings and peers relate to the child’s physical activity behaviors. In line with this, in an umbrella systematic review, probable evidence was found that having a companion for physical activity and receiving encouragement from significant others was associated with higher physical activity in both children and adolescents [44]. However, with the advancing age of the child, it is likely that parental influences on children’s physical activity behavior can change [45]. 

Although there has been a focus on the associations between PAPP and children’s physical activity, very little evidence is available regarding moderators of parental influences on children’s activity behavior [40]. To date, relatively little is known about the factors explaining PAPP itself [38] and more emphasis should be given to these aspects in parental support programs. Although no significant associations were found in this study, the conceptual model may prove useful for future research and also highlight the relevance of considering the children’s activity style when developing family-based interventions for increasing physical activity among children.

In this study, we tested a new conceptual model using SEM between PAPP and children’s physical activity and sedentary behavior assessed by accelerometry, with the inclusion of children’s activity style as a moderating factor. It was a strength of the study that we used the PAPP domains that have recently been updated and have shown good psychometric properties [33]. A limitation to the items used to assess the children’s activity style [24] was that only three out of five items were used in this study, although this decision was made to keep the response burden low. The main limitation of this study was the relatively small sample size, which only allows cautious interpretations of results. Shi et al. highlighted that a sample size of *n* = 200 provides a reasonable estimate of CFI in models with 30 or a smaller number of observed variables [46]. The authors also advised caution in interpreting RMSEA for small models with 10 or less observed variables [46]. Therefore, the CFI was interpreted cautiously to assess the model fit, with 88 complete cases included in the analysis and over 10 observed variables. The rather large model that was estimated contained many parameters, which decreased the degrees of freedom. Although the model fit was acceptable when looking at the indices RMSEA and chi-square/degrees of freedom ratio, no associations reached statistical significance; therefore, the estimates must be interpreted cautiously. Furthermore, due to the small sample size, no comparisons between mothers and fathers and boys and girls were possible, which would have been interesting to explore and can be of interest in future studies. When using the results from this study in other settings than the one targeted in this manuscript, the generalizability needs to be considered.

## 5. Conclusions

The current study could not confirm the hypothesis that the PAPP domains Structure, Neglect/control, or Autonomy promotion were significantly associated with children’s activity indicators when the moderating effect of children’s activity style was taken into consideration. However, it was confirmed that the conceptual model was reasonable to describe the data with the activity style of the child as a moderator between PAPP and the child’s physical activity. This study suggests that children’s activity style should be considered in future, larger studies to better understand the relationships between PAPP and children’s physical activity, and these results can help guide such studies. Ultimately, a better understanding of this relationship may lead to more effective family-based interventions with a focus on promoting children’s physical activity.

## Figures and Tables

**Figure 1 children-09-00248-f001:**
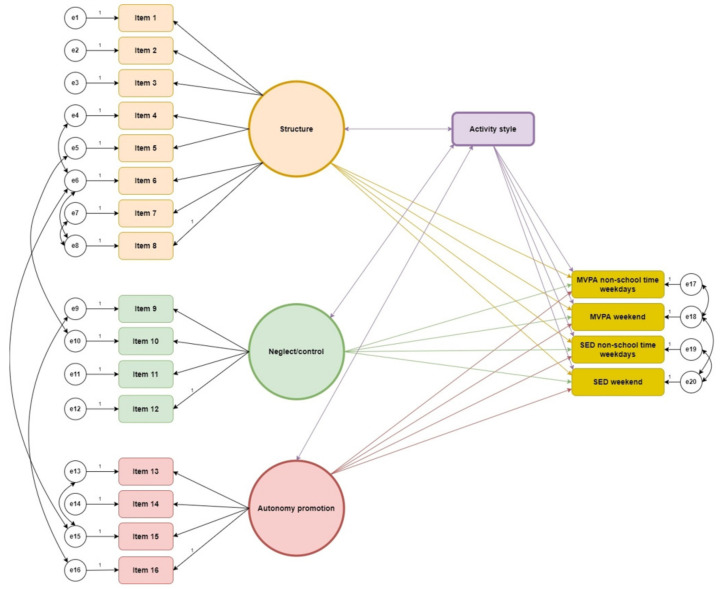
The conceptual model with children’s activity style as a moderator of the relationship between the latent PAPP domains Structure, Neglect/control, and Autonomy promotion and children’s physical activity.

**Table 1 children-09-00248-t001:** Physical activity parenting practices (PAPP) domains, constructs, and items included in the study from the item bank by Masse et al, which was published under the Creative Commons Attribution 4.0 International License [33].

Domain	Constructs	Items					
Structure	Nondirective support	1. Participate in any physical activity (such as playing ball or sports) with your child					
		2. Ask your child to exercise or be physically active with you					
		3. Walk or bike with your child to go to places that are near your home(a few minutes away) even though it would be quicker to drive					
		4. Keep track (in your head or writing down) whether your child did 60 min of physical activity or exercise every day					
		5. I talk about my physical activity with my child					
	Supportive expectation	6. I make sure my child has the physical activity or sport equipment to use when they want to play outside (such as soccer balls, basketballs, or active outdoor toys)					
	Restrict inside PA	7. How often do you restrict active play (e.g., ball games, running, and wrestling) inside your home					
		8. How often do you prevent your child from playing actively for fear of someone getting hurt					
Neglect/control	Coercive control	9. I have to nag or constantly remind my child to be physically active in their free time					
		10. I threaten to take away privileges (e.g., TV or video game times) if my child does not spend time being physically active in their free time					
		11. My child knows that I get upset and angry at them if they are not participating in physical activity in their free time					
		12. To encourage my child to be physically active, I promise a sweet or salty treat (e.g., dessert) if they are active					
Autonomy promotion	Autonomy support	13. Praise your child for being physically active or for participating in sports or physical activity classes					
		14. On the weekends, I encourage my child to play outside when the weather allows					
	Guided Choice	15. I provided my child with choices about the physical activity they do					
		16. When I discuss with my child when they should be active, we can quickly agree on a solution we are both happy with					

**Table 2 children-09-00248-t002:** Descriptive characteristics of the participating children and their parents.

				Total
				*n* = 88
				Mean (SD)
Girls				51.1%
Mothers responded				78.4%
Family education level-low ^a^				35.2%
Family born outside the Nordic region ^b^				31.8%
Age (years)				6.3 (0.3)
Children with underweight ^c^				5.8%
Children with normal weight ^c^				71.3%
Children with overweight ^c^				14.9%
Children with obesity ^c^				8.1%
Activity style (0–5)				3.6 (0.8)
Physical activity parenting practices domains				
Structure (0–40)				27.3 (4.2)
Neglect/Control (0–20)				17.3 (2.9)
Autonomy promotion (0–20)				15.2 (2.5)
Physical activity				
MVPA, weekend				60 (24)
MVPA, non-school time weekdays				22 (9)
Sedentary, weekend				416 (63)
Sedentary, non-school time weekdays				167 (36)

MVPA = moderate to vigorous physical activity. ^a^ Lowest reported education level within a family, defined as ≤12 years. ^b^ Defined as one or both parents born outside the Nordic region. ^c^ Defined according to IOTF cutoffs [30].

**Table 3 children-09-00248-t003:** Values for fit indices for the three different models.

Model	χ2/df	CFI	RMSEA (Lower;Upper)	SRMR
1	1.968	0.569	0.106 (0.088;0.123)	0.131
2	1.905	0.579	0.102 (0.085;0.119)	0.127
3	1.338	0.851	0.062 (0.038;0.083)	0.110

χ2/df = chi-square/degrees of freedom, CFI = comparative fit index, RMSEA = root mean squared error of approximation, SRMR = standardized root mean residual.

**Table 4 children-09-00248-t004:** The relationships between the three latent PAPP domains and children’s physical activity in Model 1 without activity style and Model 2 including activity style as a moderator.

						b	se	*p*	β
**Model 1**									
Direct effects									
Structure		on	MVPA non-school time weekdays			17.32	24.22	0.47	0.16
			MVPA weekend			67.32	87.55	0.44	0.25
			SED non-school time weekdays			49.22	76.12	0.52	0.12
			SED weekend			42.73	96.83	0.66	−0.06
Neglect/control		on	MVPA non-school time weekdays			4.00	3.95	0.31	0.12
			MVPA weekend			1.02	9.95	0.92	0.01
			SED non-school time weekdays			4.27	15.36	0.78	0.03
			SED weekend			12.33	26.85	0.65	−0.06
Autonomy promotion		on	MVPA non-school time weekdays			6.65	6.47	0.30	0.14
			MVPA weekend			12.09	15.56	0.44	−0.10
			SED non-school time weekdays			15.24	23.38	0.52	−0.08
			SED weekend			−7.71	38.88	0.84	−0.03
**Model 2**									
Direct effects									
Structure		on	MVPA non-school time weekdays			14.00	20.47	0.49	0.13
			MVPA weekend			63.61	82.56	0.44	0.24
			SED non-school time weekdays			50.33	76.30	0.51	0.12
			SED weekend			49.90	100.83	0.62	−0.07
Neglect/control		on	MVPA non-school time weekdays			2.86	4.24	0.50	0.09
			MVPA weekend			−1.05	10.97	0.92	−0.01
			SED non-school time weekdays			11.70	16.93	0.49	0.09
			SED weekend			−4.73	29.44	0.87	−0.02
Autonomy promotion		on	MVPA non-school time weekdays			5.74	4.54	0.21	0.20
			MVPA weekend			−8.39	11.19	0.45	−0.11
			SED non-school time weekdays			−4.34	16.50	0.79	−0.04
			SED weekend			10.00	29.02	0.73	0.05

Coefficients: b = unstandardized and β = standardized.

**Table 5 children-09-00248-t005:** Relationships between the three latent PAPP domains and children’s physical activity and activity style in Model 3.

						b	se	*p*	β
Direct Effects									
Structure		on	MVPA non-school time weekdays			18.17	28.99	0.53	0.14
			MVPA weekend			72.30	107.87	0.50	0.22
			SED non-school time weekdays			77.39	121.70	0.53	0.16
			SED weekend			51.60	119.68	0.67	−0.06
Neglect/control		on	MVPA non-school time weekdays			2.43	3.87	0.53	0.08
			MVPA weekend			−1.90	10.02	0.85	−0.02
			SED non-school time weekdays			10.17	15.39	0.51	0.08
			SED weekend			−1.81	26.96	0.95	−0.01
Autonomy promotion		on	MVPA non-school time weekdays			4.83	3.22	0.13	0.17
			MVPA weekend			−1.38	7.92	0.86	−0.02
			SED non-school time weekdays			12.51	12.36	0.31	−0.11
			SED weekend			14.88	21.54	0.49	0.08

Coefficients: b = unstandardized and β = standardized.

## Data Availability

The datasets are not available for download in order to protect the confidentiality of the participants. The data are held at Karolinska Institutet.

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
