# Peer review of "Relationships between Physical Activity Parenting Practices and Children’s Activity Measured by Accelerometry with Children’s Activity Style as a Moderator—A Cross Sectional Study"

_children, 2022, doi:10.3390/children9020248_

Round 1
Reviewer 1 Report
Attached.

Reviewer 2 Report
First question:
What was the reason that only girls were included in this research?
Second question: Do you think that such a small number of subjects, i.e. 88, is significant for this research? or can it be a limit of the research?
Material and methods
Did all subjects have accelerometers for this research or was there another way to use this device? also, what happened to the accelerometers at night and when the kids took a shower ... please detail
Otherwise the methodology and the results are very clearly presented.
As the authors say ''The main limitation was the sample size'' .... why the authors not choose a larger number of subjects? I think it would have been very interesting to see this, and also the participation of the boys would have been interesting for a comparative analysis by gender.
Reviewer 3 Report
Abstract:
Point 1:
Line 20: Study participants were 88 children (average age 6.3 years, SD 0.3), Please write the age standard deviation together.
Point 2:
Line 26-27: No significant associations between the PAPPs Structure, Neglect/control and Autonomy promotion and measures of physical activity were found. Please indicate specific results.
Introduction:
Point 3:
Line 49-50: Another serious public health concern is the increase of child obesity during the last decades [8-10]. Prevention and treatment early in life should remain a priority [11-14]. Please don't make a single sentence into a paragraph.
Materials and Methods
Point 4:
Line138-139: The family was classified as “high level of education”, if at least one parent reported an education longer than 12 years, otherwise the family was classified as “low level of education”. This statement is not sufficient and lacks basis.
Point 5:
Line157-158: All items were translated into Swedish and back translated by the research team and pre-tested for comprehension in parents with pre-school children. Please don't make a single sentence into a paragraph.
Point 6:
Line191-192: All questionnaires were web-based, in Swedish, and accessed by parents via the project website. Please don't make a single sentence into a paragraph.
Point 7:
Line215-217: Model fit for RMSEA: < 0.05 indicates good model fit, < 0.08 acceptable fit, and 0.08 to 0.10 mediocre fit. Model fit for CFI: ≥0.9 indicates acceptable fit, and 0.95 good fit. Model fit for SRMR: <0.1 indicates acceptable fit [37]. The indicators are not comprehensive when evaluating the fitting of the model.
Results
Point 8:
Line223-228: Table 2 presents the descriptive characteristics for the included children and the parents. In total, 88 parents (69 mothers, 19 fathers) and 88 children were included. Of the parents, 35% had finished 12 years or less of schooling and 32% were born outside the Nordic region. The children were on average 6.3 (SD 0.3) years old; 49% were boys (n=43) and the rate of overweight and obesity among the children was 23%. Overall, the children spent on average 60 minutes (SD 2.4) in MVPA during weekends and 22 minutes (SD 9.4) during non-school time on weekdays. Please write the mean and standard deviation together.
Point 9:
Line255-256: Coefficients of table 4 need to be annotated.
Point 10:
Line233: lower (32% vs 59%, p<0.001). Please be revised it, p – p.
Point 11:
Line255: Please be revised it. “P-value” —— P. Please revise the following uniformly.
Round 2
Reviewer 1 Report
None
Reviewer 3 Report
Good revision.